# Stabilization of Pickering Emulsions by Hairy Nanoparticles Bearing Polyanions

**DOI:** 10.3390/polym11050816

**Published:** 2019-05-07

**Authors:** Ying Zhang, Kaimin Chen, Lan Cao, Kai Li, Qiaoling Wang, Enyu Fu, Xuhong Guo

**Affiliations:** 1State Key Laboratory of Chemical Engineering, School of Chemical Engineering, East China University of Science and Technology, Shanghai 200237, China; zy12fearless@163.com (Y.Z.); lancaoecust@outlook.com (L.C.); 2College of Chemistry and Chemical Engineering, Shanghai University of Engineering Science, Shanghai 201620, China; 18301939658@163.com (K.L.); woodsues@outlook.com (Q.W.); fuenyu1234@163.com (E.F.)

**Keywords:** hairy nanoparticles, pickering emulsion, photoemulsion polymerization

## Abstract

Pickering emulsions are increasingly applied in drug delivery, oil–water separation, composite materials preparation, and other fields. However, systematic studies on the stabilization of Pickering emulsions to satisfy the growing application demands in multiple fields with long-term conservation are rare. Compared to conventional solid nanoparticles, polyanion-modified hairy nanoparticles are more stable in practical environments and are investigated in this study. Poly (sodium *p*-styrenesulfonate) was grafted to a polystyrene (PS) core via a photoemulsion polymerization. A hairy nanoparticle bearing polyanions called poly (sodium *p*-styrenesulfonate) brush (PS@PSS) was synthesized. The size and uniformity of the Pickering emulsions stabilized by PS@PSS were investigated via a polarizing microscope. The stability of Pickering emulsions were optimized by adjusting critical factors like ultrasonic power and time, standing time, oil phases, salt concentration, and water:oil ratio. Results indicated that the Pickering emulsions could be stabilized by PS@PSS nanoparticles, which showed remarkable and adjustable partial wetting properties. It was found that the optimized conditions were ultrasonic power of 150 W, ultrasonic time of 3 min, salt concentration of 0.1 mM, oil phase of hexadecane, and water:oil ratio of 1:1. The formation and stability of Pickering emulsion are closely related to the hairy poly (sodium *p*-styrenesulfonate) brush layer on the nanoparticle surface.

## 1. Introduction

Conventional emulsions used in daily life are commonly stabilized by low molecular weight surfactants that are thermodynamically stable. However, this kind of emulsion increases the costs and environmental impact [1]. Pickering emulsions are among the most well-studied alternative strategies; these were first proposed by S.U. Pickering [2] and are applied in various fields such as drug delivery [3,4], cosmetics [5,6], food science [7], manufacturing of microcapsules [8,9], and porous materials [10,11,12,13]. Pickering emulsions not only have outstanding stability, but also satisfy the demand for intelligent stimuli-responsive emulsions because of functional nanoparticles anchored at the water/oil interface such as spherical nanoparticles [14], nanocrystals [15,16], and nanotubes [17]. Saigal et al. [18] reported that thermally-responsive emulsions could be created with the SiO_2_-PDMAEMA particles such that stable emulsions prepared at low temperature were rapidly broken by increasing the temperature above the critical flocculation temperature (CFT). Chen et al. prepared a new class of donor-acceptor Stenhouse adduct (DASA)-functionalized silica microspheres to formulate visible light-controlled Pickering emulsions. This unique inversion behavior was applied to control the encapsulation and release of fluorescein sodium salt [19]. In recent years, studies have focused on the functional applications of Pickering emulsions stabilized by creative synthetic polymeric nanoparticles, but few have evaluated the stabilization of Pickering emulsions—this is significant for their practical applications.

For Pickering emulsion stabilization, several factors must be considered including particles (types, size, aspect ratio, and grafting density), salt concentration, and oil phase. For instance, Madivala et al. [20] investigated the effect of particle aspect ratio on the stability of Pickering emulsions. They reported spindle-type hematite particles with higher aspect ratio particles deposited more readily at the water/oil interface. This demonstrated that destabilization can be achieved simply by shape changes. Katepalli et al. [21] found that fumed silica particles could easily form stable volume-filling networks under high salt concentration where the interparticle interactions were attractive. Tsuji et al. [22] prepared oil-in-water (O/W)-type Pickering emulsions stabilized by PS@PNIPAM hairy particles from various oils (eg. heptane, hexadecane, and toluene), but emulsions could not be formed when 1-undecanol was used as the oil phase because the wettability of hairy particles was higher for 1-undecanol than for water.

Nevertheless, due to complicated operation processes and repeated experimental procedures which cause vast time expenditure and material waste [23], there are relatively few systematic studies on the stability of Pickering emulsions to satisfy the growing application demands in multiple fields with long-term conservation. Hairy nanoparticles bearing polyelectrolytes, composed of a core and a layer of polymer chains densely grafted via covalent bonds on the core surface [24], are the most used candidates for preparation of Pickering emulsion [25,26,27,28] and further applied to achieve the tunability of emulsification and demulsification. Polyanion-modified nanoparticles are becoming an appealing choice due to many advantages like feasibility, uniform size, and outstanding stability of products in a practical environment. In this work, poly (sodium *p*-styrenesulfonate)-modified polystyrene nanoparticles (PS@PSS) were synthesized via surface-initiated photoemulsion polymerization [29] and were used as the solid surfactant to prepare Pickering emulsions under various conditions. The effects of ultrasonic power, ultrasonic time, standing time, oil phases, salt concentration, and water:oil ratio on the formation and stabilization of Pickering emulsions were systematically investigated to establish the optimum condition for the formation of the desired Pickering emulsions.

## 2. Materials and Methods

### 2.1. Materials

Styrene was purchased from Lingfeng Chemical Reagent Co., Ltd. (Shanghai, China). Sodium dodecyl sulfate (SDS) was supplied by Amresco (Shanghai, China). K_2_S_2_O_8_ (KPS, 99%), hexadecyltrimethylammonium bromide (99%), divinylbenzene (m- and p-mixture, 55% DVB in ethylene ethylbenzene (EVB) and diethyl benzene (DEB), stabilized with tert-butylcatechol (TBC)) were purchased from J & K Chemical (Shanghai, China). Sodium *p*-styrenesulfonate (95%) was purchased from Jiuding Chemical Reagent Co., Ltd. (Shanghai, China). The n-hexadecane (98%) was purchased from Shanghai Aladdin Industrial Co., Ltd. (Shanghai, China). Ethanol (99.7%), acetone (99.5%) and sodium chloride (99.5%) were purchased from Shanghai Titan Scientific Co., Ltd. (Shanghai, China). The n-decane (95%) was purchased from Shanghai Chaocong Chemical Co., Ltd. (Shanghai, China).

Styrene was stored in a refrigerator at 4 °C after removing the polymerization inhibitor via reduced pressure distillation. The homemade and high-efficiency photo initiator 2-[p-(2-hydroxy-2-methylpropiophenone)]-ethyleneglycolmethacrylate (HMEM) was synthesized as described in our previous work [30]. The water used for all experiments was purified by reverse osmosis (Millipore Milli-RO, Merck KGaA, Darmstadt, Germany) and subsequent ion exchange (Millipore Milli-Q, Merck KGaA, Darmstadt, Germany). All others were used as received.

### 2.2. Synthesis of PS@PSS Hairy Nanoparticles

#### 2.2.1. Synthesis of Crosslinked Polystyrene (PS) Core Emulsion

PS core emulsion was synthesized by emulsion polymerization as follows. Firstly, 0.46 g KPS and 0.14 g SDS were dissolved in 200 mL H_2_O, afterwards solutions were transferred into a 250 mL three-necked and round-bottomed flask with mechanical stirring. Afterwards, 8 g styrene and 0.27 g DVB (1/30 of styrene in mass) were added into the flask. After being protected by a nitrogen atmosphere, the whole system was heated to 80 °C with an oil bath at 300 rpm for 2 h. Secondly, the reactor temperature was set as 70 °C. Once the temperature reached 70 °C, 8 g HMEM acetone solution (1:9 *w*/*w*) was added dropwise into the flask at “starved condition”. After adding HMEM, the reaction continued for 2.5 h, maintaining the temperature at 70 °C and mechanical stirring with 300 rpm. Thirdly, PS core was cooled to ambient temperature and was purified by ultrafiltration using deionized water until the conductivity was constant.

#### 2.2.2. Synthesis of Hairy Nanoparticles

PS@PSS was synthesized by photoemulsion polymerization as reported previously [29]. This method was widely applied to synthesize hairy particles in our previous work and the grafting density of the surface layer could only be adjusted within certain limits [31]. The grafting density of PS@PSS nanoparticles was near 0.05 cm^−2^ by calculation model [32]. Typically, a mixture of 30 g PS core emulsion and 1.84 g sodium *p*-styrenesulfonate were added into a home-made photoreactor (Appendix A) and the reaction volume was controlled as 150 mL with supplementary H_2_O. After being protected by nitrogen, the reaction was carried out for 2.5 h with a magnetic stirrer under UV light at 25 °C. When the synthesis was accomplished, PS@PSS was transferred into a dialysis bag for further purification. The concentration of PS@PSS was 1.02 wt.%. Figure 1 shows the structures of obtained PS@PSS hairy nanoparticles.

### 2.3. Preparation of Pickering Emulsion

Pickering emulsions are prepared with oil phase, water phase, and solid particles. Here decane or hexadecane were selected as the oil phase. Taking decane and PS@PSS as an example, in order to obtain homogeneous emulsion, PS@PSS and decane were added in a centrifuge tube at a certain volume ratio (e.g., 1:1) and the tube was fixed on the clamp in the ultrasonic homogenizer (25 kHz, SCIENTZ-IID, Ningbo Scientz Biotechnology Co., Ltd.) with a 6 mm diameter titanium probe (Appendix A). The probe must be immersed into the mixture in the tube. The ultrasonic power (W) and ultrasonic time (min) were set as a value with power variation between 100 W and 300 W during 1 min or 5 min. The duration of ultrasonication was 2 s per time and the interval time was also 2 s, in support of heat dissipation.

### 2.4. Characterization

#### 2.4.1. Dynamic Light Scattering (DLS)

The size of solid particles synthesized was measured by PSS Nicomp 380 particle sizing systems (Santa Barbara, CA, USA) at 25 °C. Samples were diluted to 0.01 wt.% with pure water when scattering intensity was around 300 kcps. Intensity-average size and Gaussian polydispersity index (PDI) were obtained for analysis.

#### 2.4.2. Transmission Electron Microscopy (TEM)

TEM were conducted on a JEM-1400 electron microscope equipped with a CCD camera (JEOL Japan, Tokyo, Japan). Samples were prepared by drying diluted droplets on carbon-film coated copper grids at room temperature and the CCD camera was inserted in for further observations.

#### 2.4.3. Polarizing Microscope Observation

One droplet of Pickering emulsion was placed on a microscope slide and covered with a coverslip at room temperature. The samples were observed by Leica Microscope DM 2500P (Wetzlar, Germany) connected with a digital camera (Gatan 830 CCD, CA, USA) under ambient conditions, in order to estimate the size and distribution of Pickering emulsion droplets. Pickering emulsion images were obtained via a digital camera which accompanied the microscope. Furthermore, Pickering emulsion size and distribution were measured and analyzed based on the images by Nano Measurer 1.2.5 Software (Shanghai, China). The selected area of Pickering emulsion droplets was 120 × 90 μm.

#### 2.4.4. Contact Angle Analysis

The partial wetting properties of PS@PSS nanoparticles were investigated by contact angle analysis. The PS@PSS aqueous solution was tiled on the clean glass plate and dried by oven. A DI water droplet was contacted by the plate and images were taken continuously using a CCD camera (Leica DFC 9000, Wetzlar, Germany) attached to the Contact Angle Measuring Device (JC2000D, Shanghai Zhongchen Co., Ltd., Shanghai, China). Therefore, the contact angle was calculated by fitting the curve. Three measurements were carried out and an average was obtained as the result.

## 3. Results and Discussion

### 3.1. Synthesis of PS@PSS Hairy Nanoparticles

Dynamic light scattering (DLS) and transmission electron microscopy (TEM) were used for size analysis of polystyrene (PS) core emulsion and subsequent hairy nanoparticles. First, DLS that could measure the hydrodynamic diameter (Rh) was applied to demonstrate the size increase from PS core to PS@PSS in an aqueous solution. Table 1 shows that the size of PS core used for preparing PS@PSS was 74.2 nm. After photoemulsion polymerization, the size of the PS@PSS was 252.8 nm, and the PDI was 0.018. The PS@PSS nanoparticles had narrow diameter distribution with solid content of 1.02 wt.%.

TEM was used to study the size and morphology of these nanoparticles (Figure 2). The PS core size was around 70 nm (Figure 2a) and the PS@PSS size was around 190 nm (Figure 2b). Thus, the thickness of the PSS chain was about 60 nm (Figure 2b), indicating that the PSS chains were successfully grafted onto the PS core. In general, the PS core size agreed with the DLS data, but the PS@PSS size was much smaller than 252.8 nm (DLS data). This was because the PSS chains would shrink and the size would decrease during the drying process of TEM sample preparation; the edge of the PSS chains was also too sparse to be observed easily in Figure 2b. In addition, DLS gives the hydrodynamic diameter (Rh), which considers the sphere of hydration. This made the apparent size (252.8 nm) larger than that from TEM (~190 nm).

### 3.2. Ultrasonic Power and Time

Ultrasonication is the most frequently used method to apply mechanical forces to synthesize Pickering emulsions. Sonication is also effective in Pickering emulsion preparation because it can simultaneously emulsify and force particle adsorption onto droplet interfaces [33]. Thus, ultrasonic conditions determined whether Pickering emulsion was well prepared or not. We chose ultrasonic power and time as the first two factors because they were key parameters of ultrasonic conditions.

Figure 3a shows microscope images of hexadecane/H_2_O Pickering emulsion stabilized by PS@PSS nanoparticles under different ultrasonic powers (100 W, 150 W, 200 W, and 300 W). The droplet size and number distribution were measured and analyzed via Nano Measurer Software (Figure 3b). In this study, all the Pickering emulsions were O/W emulsions due to the hydrophilicity of the hairy nanoparticles. Microscopic observations (Figure 3a) revealed that the droplets size of hexadecane/H_2_O Pickering emulsions stabilized by PS@PSS were the most uniform at 150 W. If the ultrasonic power decreased to 100 W, then the emulsion droplets became inhomogeneous and the droplet amount was relatively small (Figure 3b). If the ultrasonic power increased to 200 W, then the emulsion droplet uniformity was also worse than that under 150 W and the average size decreased to 6.2 μm. The size analysis in Figure 3b indicated that some droplets became obviously larger and small size droplets came out. Once the ultrasonic power increased to 300 W, Pickering emulsion droplets were much more inhomogeneous and the polarization of the droplet size could not be ignored. Clusters with relatively small particles were observed in the oil phase of a single Pickering emulsion droplet. Thus, the results showed that the distribution of emulsion droplet size was ideally normal around 7.5 μm when the ultrasonic power was 150 W; this concurred with the microscope images. This demonstrated that the Pickering emulsion was the most stable. Lower ultrasonic power could not form complete emulsions while higher ultrasonic power disturbed the preparation of Pickering emulsions. Costa et al. [34] found that zeta potential of chitosan nanoparticles gradually decreased and Pickering emulsion droplets became more homogeneous when ultrasonic power was tuned from 100 W to 450 W, but more drastic treatments (600 W) were accompanied by an increase in zeta potential.

Ultrasonic power can impact the diameter of PS@PSS nanoparticles. Costa et al. [34] explored particle size distribution of chitosan particles untreated or treated with different ultrasonication power. They found that higher ultrasonication power reduced the particle size. We then investigated the effects of ultrasonic power on PS@PSS nanoparticle size. Table 2 shows the size of PS@PSS by DLS after treatment with an ultrasonic homogenizer under different powers for 3 min. The thickness of PS@PSS decreased markedly at 200 W. The PSS chains of PS@PSS could cleave, and the hydrophilicity of PS@PSS weakened. Thus, larger Pickering emulsion droplets were produced at 200 W. In general, the PS@PSS size decreased with increasing ultrasonic power. Figure 3c shows that the contact angle of PS@PSS increased with ultrasonic power. It was demonstrated that ultrasonic power gave rise to wettability of PS@PSS nanoparticles, resulting in weaker hydrophilicity. This led to an increase in the Pickering emulsion droplet size.

The particle migration may cause the decrease in Pickering emulsion droplets when the ultrasonic power rose from 150 W to 200 W. Low power could not provide sufficient energy to “transport” the PS@PSS nanoparticles to the interface of hexadecane and water phase; thus, ultrasonic power (100 W) was too low to form complete Pickering emulsion, and the well-dispersed situation could not reached. High power could affect the droplet size and the stability of Pickering emulsion. When the ultrasonic power increased from 100 W to 150 W, the ultrasonic energy strongly pushed PS@PSS that existed in water phase previously to the interface of water and hexadecane. After withdrawal of the homogenizer, the PS@PSS anchored at the interface was thermodynamically stable. When the ultrasonic power continued to increase up to 200 W, Pickering emulsion droplet size became larger and inhomogeneous. This effect likely occurred because the greater ultrasonic energy created partial PS@PSS that had anchored at the water/hexadecane interface migrated back into the water phase. This is similar to the breaking of a conventional emulsion by mechanical stirring.

A special emulsion was formed when the ultrasonic power increased up to 300 W. The image shows some inhomogeneous clustering in the oil phase of the Pickering emulsion droplets. There were two reasons that could explain this phenomenon. First, higher ultrasonic power further broke the polymer chains of partial PS@PSS, and the particle size became smaller. This made some emulsion droplets become larger. Second, the ultrasonic power (300 W) was so strong that the hairy polyelectrolytes in some PS@PSS decreased dramatically from 252 nm to 236 nm as shown in Table 2. This increased the hydrophobicity and made them aggregate and migrate into the inner oil phase.

Sonication time was another vital factor of Pickering emulsion preparation. As shown in Figure 4, the system consisting of oil phase, water phase, and PS@PSS varies with ultrasonic time. When the sample was homogenized for only one minute, there was a layer of oil phase floating on the Pickering emulsion (inset in Figure 4a). The images also illustrate that droplets of one minute-made emulsion were fairly rare, and the PS@PSS were not dispersed uniformly in the system (Figure 4b). Thus, both the macroscopic phenomenon and microscopic images illustrated that the ultrasonic time was too short to form Pickering emulsion completely. The Pickering emulsion droplets were uniform and homogenous when the ultrasonic time extended to 2–3 min. At longer time periods, we found that the average droplets size of emulsions shown in Figure 4c were 5.8 and 5.0 μm, respectively. This led to a slight decreasing trend. Furthermore, the quantity of emulsion droplets also decreased obviously when the ultrasonic time increased to 4 min. Even at 5 min, the PS@PSS aggregated inside the oil phase (Figure 4b). The particle size distribution shown in Figure 4c was similar with that when ultrasonic power was 300 W. One explanation for these changes in Figure 4a (1 min) was the migration of PS@PSS nanoparticles. It took some time for ultrasonic energy to migrate PS@PSS from water phase to the interface of hexadecane and water. Thus, when the ultrasonication time was 1 min, a floating oil layer existed with unstable emulsion. The ultrasonication energy from the extra 2 min was convenient and facilitated the formation of Pickering emulsions, at the same time droplets size was getting smaller, which was also observed in the previous report [35]. It was concluded that in order to stabilize the droplets, more particles were absorbed to the W/O interface with shear time. Despite the accumulation effect of ultrasonic energy on particle migration, the PS@PSS size after ultrasonic treatment under different times was analyzed by DLS (Table 3). In general, with increasing ultrasonication time, the PS@PSS size decreased somewhat even though the change was relatively small. This strengthens the PS@PSS hydrophobicity and decreases emulsion droplet size. Figure 4b (5 min) shows that a long time also led to PS@PSS aggregation and migration to the inner oil phase. It was found that PDI of PS@PSS treated for 5 min increased a lot (0.068) compared to its original PDI (0.011), which may result in part PS@PSS with a short hairy layer and loss of its hydrophilicity and finally shifted to the oil phase with a long ultrasonic treatment time. Changes in polymer structure can be created during ultrasonic treatment as a result of cavitation and research has suggested that chain cleavage could occur preferentially at weak spots in the chain [36]. PS@PSS hairy nanoparticles used in this paper is a special fuzzy structure of dense core and loosened shell, so the thickness decrease in PS@PSS shown in Table 2 happened gradually from the chains outside to the inner side [37].

Both ultrasonic power and ultrasonic time were significant factors for the formation of Pickering emulsions. Low power and short time could not provide enough ultrasonic energy for PS@PSS nanoparticles to migrate from the water phase to the water/oil interface; excessive power and long-term formation of PS@PSS aggregation broke the balance that had been formed previously. Therefore, when the ultrasonic power was 150 W and ultrasonic time was 3 min, high-quality hexadecane/H_2_O Pickering emulsions stabilized by PS@PSS were prepared and well-dispersed droplets were achieved. Here, the water:oil ratio was 1:1 and the salt concentration was 0.1 mM, which was investigated later in this study.

### 3.3. Standing Time

For Pickering emulsions prepared under optimal ultrasonic conditions, tiny changes could exist after standing for a while at ambient temperature (25 °C). Thus, we monitored the Pickering emulsions over time. Figure 5a shows microscope images of hexadecane/H_2_O Pickering emulsions stabilized by PS@PSS for one hour. Immediately after emulsion formation, droplets were rather homogeneous and the droplet size was normally distributed around 7.1 μm. As time went on, the droplets were not as homogeneous as the beginning, and average size became larger. This was up to 10.0 μm. Within 30 min, the mean size of the Pickering emulsion particles increased relatively rapidly (Figure 5b,c). During this period, the size of small droplets decreased and the large ones became larger, indicating polarization in droplets size. From 30 min to 1 h, no obvious change in emulsion particle size was observed. However, as shown in Figure 5c, the small droplets became smaller, and their number increased; the large droplet became larger, and their amount also increased. This indicated that Pickering emulsions are thermodynamically unstable; there was a ripening trend in this system. For hairy nanoparticles with partial wettability like PS@PSS, they thermodynamically preferred to anchor at the water/oil interface rather than stay in one of the bulk phases [21]. However, Ashby et al. [38] reported that the tension of particle-covered interfaces changed with time; there were various sizes of droplets. If small droplets lost oil, then compression of the adsorbed PS@PSS layer would occur with a concomitant lowering of tension. Large droplets swelled with additional oil, which may induce adsorption of PS@PSS from the bulk to cover the new interfaces. This could explain the droplet changes in one hour.

The emulsion droplet size was normally distributed, and changes were subtle indicating that polymer chains of PS@PSS nanoparticles were still relatively stable in water and PS core closely touched with hexadecane molecule with longer time. Thus, Pickering emulsions stabilized by PS@PSS nanoparticles could be conserved for a long time.

### 3.4. Oil Phase

Pickering emulsions consist of water phase, oil phase, and solid particles—these must be carefully controlled. Tsuji et al. reported that PNIPAM-carrying particles preferentially formed oil-in-water (O/W)-type Pickering emulsions with a variety of oils such as hexadecane, heptane, and trichloroethylene [22]. Meanwhile, long carbon chains of alkane are the simplest hydrophobic structures. Both decane and hexadecane have this kind of structure, and they are non-volatile, cheap, and easily available. They cannot dissolve PS@PSS hairy nanoparticles, which was verified by experiments. In spite of a series of similarities, hexadecane is more hydrophobic than decane because of six extra –CH_2_, which gives us a convenience to compare the effect of Pickering emulsion with different oil phases. Here, hexadecane/H_2_O and decane/H_2_O Pickering emulsions stabilized by PS@PSS were prepared with ultrasonic power of 150 W, ultrasonic time of 3 min, and water:oil ratio of 1:1. In the systems containing PS@PSS nanoparticles, the type of emulsions were always O/W reflecting the hydrophilic nature of this nanoparticle in oil-water systems.

From microscope images (Figure 6), we could find that hexadecane/H_2_O Pickering emulsion droplets significantly outnumbered decane/H_2_O Pickering emulsion droplets. In terms of uniformity, hexadecane/H_2_O Pickering emulsion droplets were also more homogeneous than droplets in decane/H_2_O Pickering emulsion. Furthermore, statistical results in Figure 6c revealed the obvious difference in droplets size distribution. Hexadecane/H_2_O Pickering emulsion droplets size was distributed around 4.3 μm but decane/H_2_O Pickering emulsion droplets size was much larger, which was up to 7.1 μm. This distinction resulted from the structure difference between decane and hexadecane. Compared with decane, hexadecane had extra six –CH_2_ and turned to be more hydrophobic. Accordingly, surface tensions (γ_ao_) of decane and hexadecane were 24.3 mN/m and 27.2 mN/m, respectively. Higher surface tension made hexadecane less wettable by PS@PSS nanoparticles in water phase and then the curvature of droplets would be bigger. So droplet size of hexadecane/H_2_O Pickering emulsion would be much smaller than decane/H_2_O Pickering emulsion (Figure 6a). Since the total volume of hexadecane and decane was the same, the amount of hexadecane/H_2_O Pickering emulsion droplets was larger as shown in Figure 6c.

### 3.5. Salt Concentration

It was shown that PS@PSS has no response to pH (Appendix A). But the salt concentration would affect the hydrophilicity of polyelectrolytes, and then it may consequently have an effect on formation of Pickering emulsion. Thus, we prepared hexadecane/H_2_O Pickering emulsions under various salt concentrations. The other conditions were ultrasonic power of 150 W and ultrasonic time of 3 min.

Figure 7 shows microscope images and droplets analysis of hexadecane/H_2_O Pickering emulsions stabilized by PS@PSS nanoparticles under different salt concentrations. Overall, increase in salt concentration (except 100 mM) increased the emulsion droplets from 4.7 μm to 16.7 μm (Figure 7b). The volume distribution of emulsion droplets was the narrowest when the salt concentration was 0.1 mM (Figure 7c). However, the amount of Pickering emulsion droplets did not change substantially as the salt concentration increased from 0.01 mM to 1 mM. Particularly, an emulsion could not be prepared when the salt concentration increased to 100 mM. Thus, the response of Pickering emulsions stabilized with PS@PSS to salt concentration was significant: the average size and volume of emulsion droplets showed an evident uptrend with the growth of salt concentration in Figure 7. However, there was no marked change in droplets amount when the salt concentration was adjusted.

The additional salt concentration shielded the electrostatic repulsive-force between PSS polymer chains. Then PSS chains shrank and tended to get close to the PS core. Correspondingly, the chains length became shorter and the exposed area of PS core would expand. This increased PS@PSS hydrophobicity. Therefore, emulsion droplets size and volume became larger with the increase in salt concentration. Additionally, as salt concentration increased and hydrophobicity of PS@PSS increased, some PS@PSS previously anchoring in hexadecane/H_2_O interfaces entered the hexadecane, while a few PS@PSS in water phase at first could migrate to interfaces. Thus, we could obviously observe the change in droplets size rather than droplets amount.

However, once the salt concentration became too high, no Pickering emulsion could be prepared, and the entire system flocculated (Figure 7a). Ashby et al. [38] also reported that in Pickering emulsion stabilized by clay, increasing salt concentration lead to turbid dispersions of increased viscosity. Due to the electrostatic shield effect between Na^+^ and PSS chains, the exposed area of the PS core increased sharply, and PS@PSS nanoparticles became much more hydrophobic. Thus, the nanoparticle migration rate from the hexadecane/water interface to hexadecane was higher than that from water phase to the hexadecane/water interface. PS@PSS in water phase could not exist stably and was flocculated in a mixture with hexadecane that was totally hydrophobic.

### 3.6. Water:Oil Ratio

There are two ways to control over stability and type of Pickering emulsion: tailoring the amphiphilicity of particle emulsifiers or changing the compositions of the emulsion phases [23]. Based on the Ostwald packing theory, it is well known that changing the water:oil ratio of emulsion systems in a wide range could give rise to interconversion between different types of emulsions which is commonly named emulsion phase inversion [39,40]. During the process, emulsion droplets size and the emulsion volumes also depended on the volume ratios of water and oil. As shown in Figure 8a, hexadecane/H_2_O Pickering emulsion stabilized by PS@PSS had the maximal emulsion volume when water:oil volume ratio was 1:1. When water:oil ratio was over 1, an O/W emulsion was formed successfully shown as W2O1 which meant the volume ratio of water and oil (hexadecane) was 2:1. We could clearly see that there was an obvious boundary between emulsion and water phase in Figure 8a. The lower layer was PS@PSS as verified by DLS. The size was still around 250 nm. Both the emulsions above could exist stably for three months. When water:oil ratio was less than 1, only a thin O/W emulsion layer was temporarily formed, and the emulsion was broken automatically five hours later.

A possible Pickering emulsion droplets scheme is given in Figure 8b. Moreover, microscope images in Figure 8c also showed that emulsion droplets with diameters of ~12.0 μm were the most homogeneous when the water:oil ratio was 1:1. When this ratio was over 1, Pickering emulsion phase formed at the upper layer and spherical droplets with diameters of ~8.0 μm were clearly observed in the corresponding optical microscopy images (Figure 8d).

On the one hand, no stable emulsion under low water:oil ratio was formed, which was attributed to the hydrophilic nature of PS@PSS nanoparticles. On the other hand, for stable emulsions when water:oil ratio was over 1, decreasing water:oil ratio resulted in larger emulsion droplets and more emulsion volume fractions. Zhai et al. reported that for stable emulsions with lower water contents, fewer nanoparticles were able to stabilize the droplets [41]. Therefore, there were few PS@PSS nanoparticles adsorbed at the interface of hexadecane and water to form a complete interfacial film. Then the small emulsion droplets tended to coalesce into larger ones, resulting in an increase in the uniformity of the droplet shapes.

### 3.7. PS@PSS Concentration

The concentration of PS@PSS nanoparticles is another vital factor that can affect the stabilization of Pickering emulsion. Optical images and microscope images of Pickering emulsions stabilized by PS@PSS with different concentrations were shown in Figure 9. When PS@PSS concentration varied from 1.02 wt.% to 0.34 wt.%, a reduction of emulsion fraction and an increase in hexadecane bulk phase volume were observed in Figure 9a, which was also shown in previous work [41]. Figure 9b shows that the amount of emulsion droplets had an obvious decline and the homogeneousness of droplets was getting worse with the decrease in PS@PSS concentration. In addition, the size analysis in Figure 9c indicated that the droplet size became smaller, with mean diameter from 12.0 μm to 2.2 μm. When particle concentration was quite low, there was not enough PS@PSS nanoparticle anchoring at all the water/oil interfaces. So, a layer of oil phase could be observed in Figure 9a when PS@PSS concentration was 0.34 wt.% and 0.51 wt.%, and oil phase volume became larger when particle concentration was lower (0.34 wt.%). However, when PS@PSS concentration increased to 1.02 wt.%, more particles migrated to anchor at the interface of hexadecane and water under ultrasonic treatment.

## 4. Conclusions

Compared to the articles concerning preparation of Pickering emulsions based on nanoparticles bearing hairy chains [42,43,44,45,46,47,48], this work not only focuses on the preparative conditions during the formation of Pickering emulsion, but also monitors the Pickering emulsion stability with time after preparation. It was found that both formation and stability of Pickering emulsion depend on the competition of thermodynamic and kinetic behavior during the preparative and standing process. A hairy nanoparticle bearing polyanions called poly (sodium *p*-styrenesulfonate) (PS@PSS) was applied as a solid nanoparticle to stabilize the water/oil interface during the Pickering emulsion formation. The optimized ultrasonic power is 150 W and the ultrasonic time is 3 min. It was found that low ultrasonic power or a short ultrasonic time could not provide enough ultrasonic energy for PS@PSS nanoparticles to migrate from water phase to the water/oil interface, while high ultrasonic power or a long ultrasonic time would damage the nanoparticle surface and lead to aggregation in the oil phase, which has not been reported elsewhere. Ripening of obtained Pickering emulsion occurred in 30 min, and then would be stable for three months, which is similar to what has been reported by other solid particle stabilized Pickering emulsion system [38]. The ripening could also be reduced by introducing more a hydrophobic oil phase, i.e., hexadecane, compared to a short-chain alkane like decane. Relatively homogeneous Pickering emulsion was observed at low ionic strength (0.1 mM), while heterogenous emulsion with a larger size was obtained at high ionic strength up to 10 mM. However, Pickering emulsion was hardly formed at 100 mM due to extreme hydrophobicity of the particle, a phenomenon similar to a system based on clay [38]. The optimized water:oil ratio is 1:1, and no stable emulsion was prepared when water:oil ratio was lower than 1:1 because of the hydrophilic nature of PS@PSS. Once water:oil ratio was over 1:1, lower water:oil ratio resulted in larger emulsion droplets and more emulsion volume fractions, which was also observed in hematite particle stabilized Pickering emulsions [49].

This work paves the way for Pickering emulsion formation and stabilization mechanism that Pickering emulsion equilibrium could be broken as long as the ultrasonic energy or external stimuli is large enough to migrate solid particles from interface to bulk phase. This finding not only provides the basis for preparing Pickering emulsions with superior stability and a long-term guarantee period, but also is extremely pertinent when designing Pickering emulsions for use in controlled stabilization and destabilization applications such as drug delivery and oil harvesting. Further study is needed to investigate the influence of hairy nanoparticles with different structures on the formation, microstructure, and stability of Pickering emulsion to confirm the proposed mechanism and potential application in such as controlled drug release is also under the way.

## Figures and Tables

**Figure 1 polymers-11-00816-f001:**
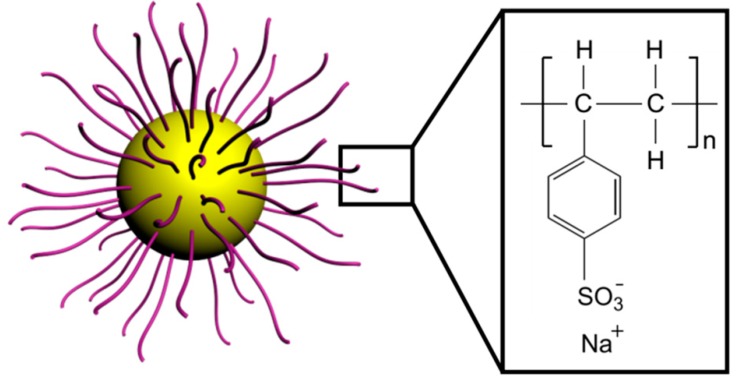
Schematic illustration of poly (sodium *p*-styrenesulfonate) (PS@PSS) structure.

**Figure 2 polymers-11-00816-f002:**
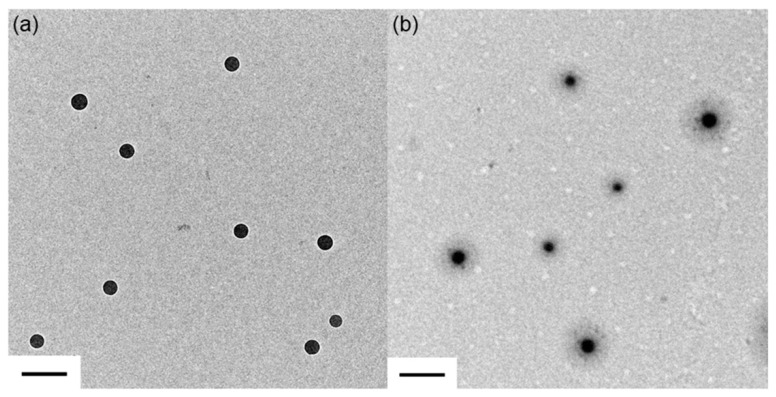
Transmission Electron Microscopy (TEM) images of (**a**) crosslinked polystyrene (PS) core; (**b**) PS@PSS hairy nanoparticles. The bar is 200 nm.

**Figure 3 polymers-11-00816-f003:**
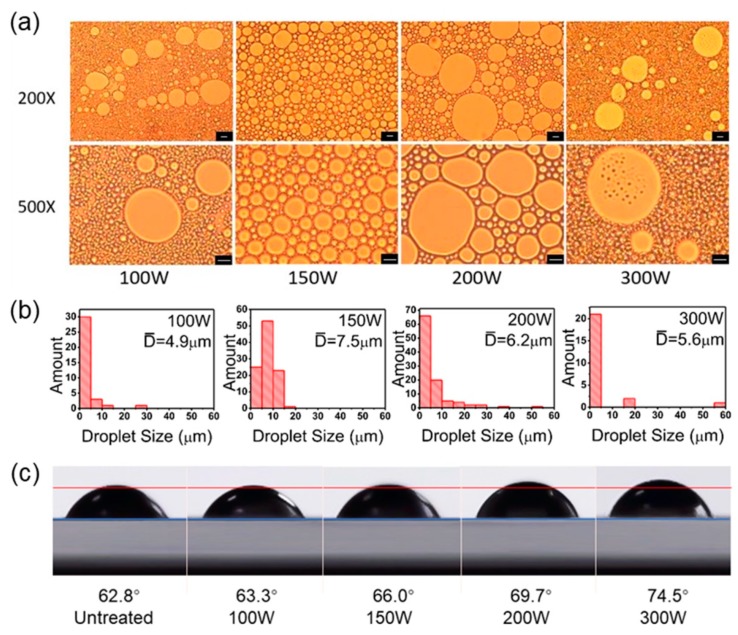
(**a**) Optical images of hexadecane/H_2_O Pickering emulsion stabilized by PS@PSS under different ultrasonic powers. Here the water:oil ratio is 1:1, salt concentration is 0.1 mM, and ultrasonic time is 3 min. The bar is 10 μm. (**b**) Pickering emulsion droplet size distribution in number corresponding to microscope images. (**c**) Contact angle images of a water droplet on the glass plate coating with PS@PSS nanoparticles and contact angle data of PS@PSS after homogenizer treatment under different powers.

**Figure 4 polymers-11-00816-f004:**
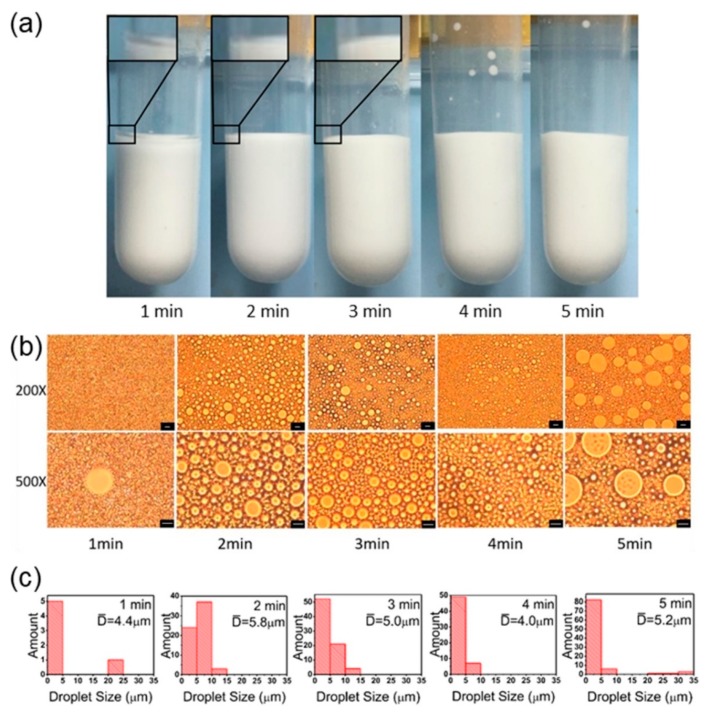
(**a**) Optical images of hexadecane/H_2_O Pickering emulsion stabilized by PS@PSS under different ultrasonic time. (**b**) Microscope images of hexadecane/H_2_O Pickering emulsion stabilized by PS@PSS under different ultrasonic time. Here the water:oil ratio is 1:1, salt concentration is 0.1 mM, and ultrasonic power is 150 W. The bar is 10 μm. (**c**) Droplets size distribution in number corresponding to microscope images.

**Figure 5 polymers-11-00816-f005:**
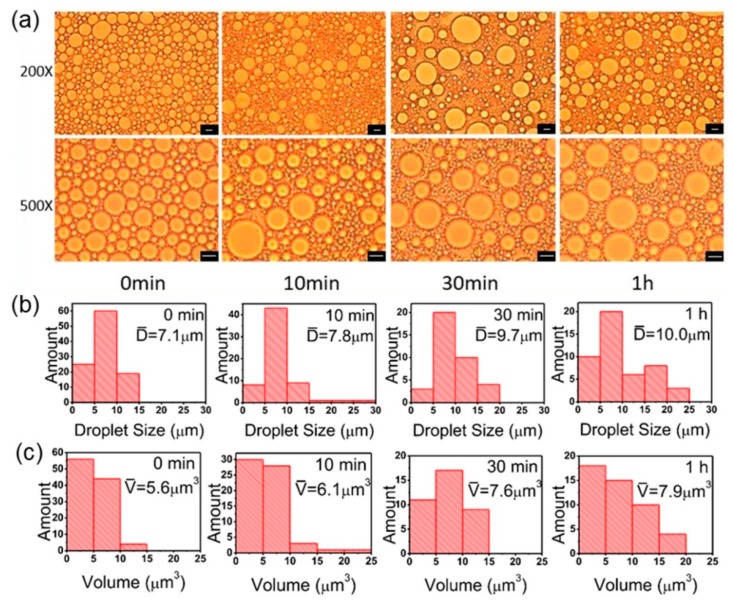
(**a**) Microscope images of hexadecane/H_2_O Pickering emulsion stabilized by PS@PSS in 1 h. Here the water:oil ratio is 1:1, salt concentration is 0.1 mM, ultrasonic power is 150 W, and ultrasonic time is 3 min. The bar is 10 μm. (**b**) Droplets size distribution in number corresponding to microscope images. (**c**) Droplets volume distribution corresponding to microscope images.

**Figure 6 polymers-11-00816-f006:**
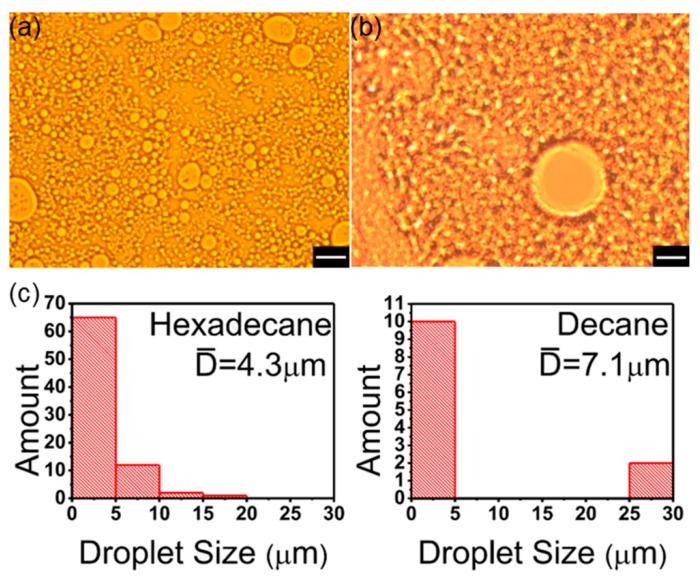
Microscope images of Pickering emulsion stabilized by PS@PSS with different oil phase (**a**) hexadecane; (**b**) decane. The magnification is 500 times. The bar is 10 μm. (**c**) Droplets size distribution in number corresponding to microscope images.

**Figure 7 polymers-11-00816-f007:**
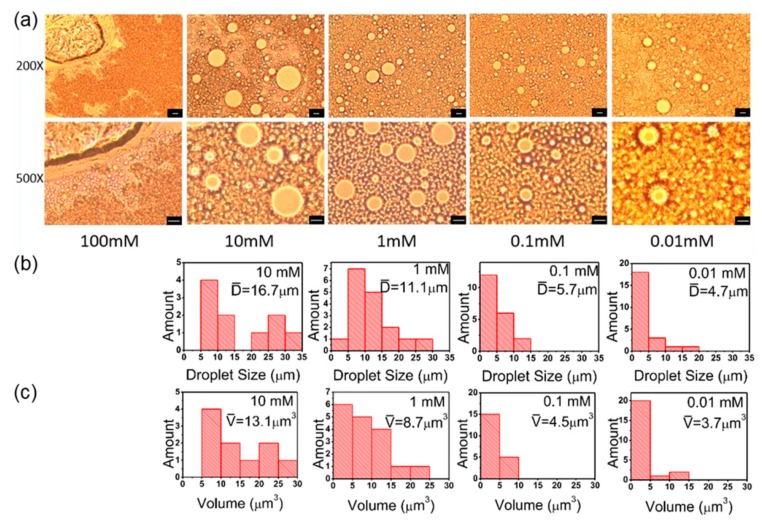
(**a**) Microscope images of hexadecane/H_2_O Pickering emulsions stabilized by PS@PSS under different salt concentrations. Here the ultrasonic power is 150 W, water:oil ratio is 1:1, and ultrasonic time is 3 min. The bar is 10 μm. (**b**) Particle size distribution in number corresponding to microscope images. (**c**) Droplets volume distribution corresponding to microscope images.

**Figure 8 polymers-11-00816-f008:**
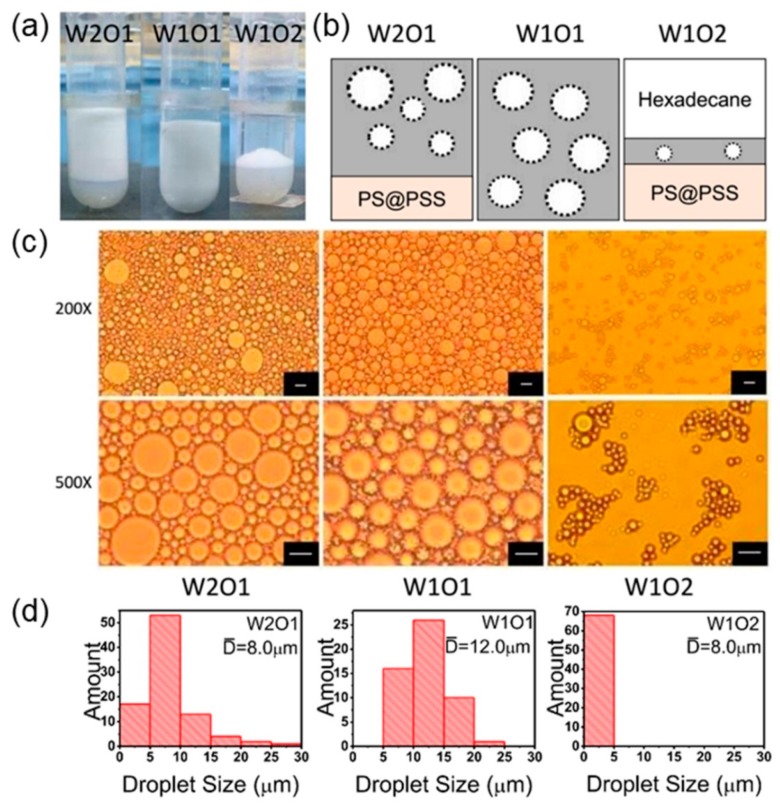
(**a**) Optical images of hexadecane/H_2_O Pickering emulsions stabilized by PS@PSS under different water:oil volume ratios after standing for 2 h; (**b**) Schematic illustrations of hexadecane/H_2_O Pickering emulsions stabilized by PS@PSS under different water:oil ratios; (**c**) Microscope images of hexadecane/H_2_O Pickering emulsion stabilized by PS@PSS under different water:oil ratios. Here the ultrasonic power is 150 W, salt concentration is 0.1 mM, and ultrasonic time is 3 min. The bar is 10 μm. (**d**) Droplets size distribution corresponding to microscope images.

**Figure 9 polymers-11-00816-f009:**
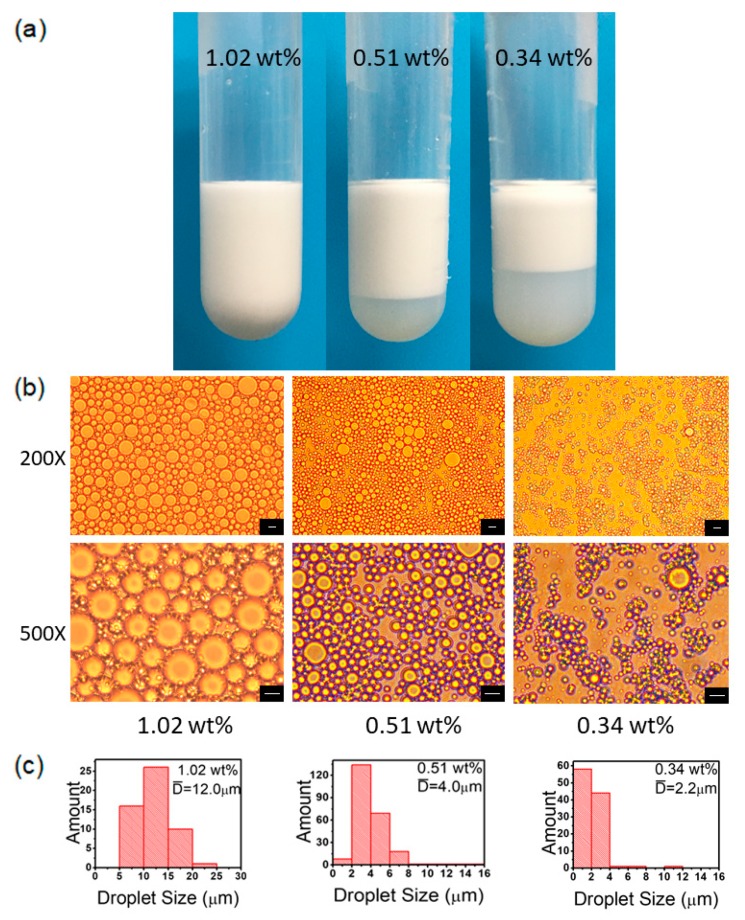
(**a**) Optical images of hexadecane/H_2_O Pickering emulsions stabilized by PS@PSS with different concentrations after standing for 10 min; (**b**) Microscope images of hexadecane/H_2_O Pickering emulsion stabilized by PS@PSS with different concentrations. Here the ultrasonic power is 150 W, salt concentration is 0.1 mM, ultrasonic time is 3 min, and water:oil ratio is 1:1. The bar is 10 μm. (**c**) Droplets size distribution in number corresponding to microscope images.

**Table 1 polymers-11-00816-t001:** Size of hairy nanoparticles by Dynamic Light Scattering (DLS).

Size and Size Distribution	PS Core	PS@PSS	Thickness (nm)
Diameter (nm) ^1^	74.2 ± 0.3	252.8 ± 1.2	89.3 ± 0.7
PDI ^1^	0.001 ± 0.001	0.018 ± 0.009	-

^1^ Measured by DLS and took the average as the result.

**Table 2 polymers-11-00816-t002:** Size of PS@PSS after homogenizer treatment under different powers.

Entry	Diameter (nm) ^1^	Thickness (nm) ^2^	PDI ^1^
PS@PSS (untreated)	252 ± 1.2	89 ± 0.8	0.018 ± 0.009
PS@PSS (100 W)	249 ± 0.9	88 ± 0.6	0.027 ± 0.013
PS@PSS (150 W)	247 ± 1.4	86 ± 0.9	0.022 ± 0.008
PS@PSS (200 W)	239 ± 2.7	82 ± 1.5	0.056 ± 0.021
PS@PSS (300 W)	236 ± 1.8	81 ± 1.1	0.037 ± 0.015

^1^ Measured by DLS and took the average as the result. ^2^ Calculated using PS@PSS diameter minus PS core diameter and then divided by two.

**Table 3 polymers-11-00816-t003:** Size of PS@PSS after homogenizer treatment under different time.

Entry	Diameter (nm) ^1^	Thickness (nm) ^2^	PDI ^1^
PS@PSS (1 min)	248 ± 0.9	87 ± 0.6	0.011 ± 0.002
PS@PSS (2 min)	246 ± 0.4	86 ± 0.4	0.032 ± 0.013
PS@PSS (3 min)	243 ± 0.4	85 ± 0.4	0.027 ± 0.010
PS@PSS (4 min)	242 ± 0.3	84 ± 0.3	0.047 ± 0.018
PS@PSS (5 min)	241 ± 0.2	83 ± 0.3	0.068 ± 0.024

^1^ Measured by DLS and took the average as the result. ^2^ Calculated using PS@PSS diameter minus PS core diameter and then divided by two.

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
