# Peer review of "Stabilization of Pickering Emulsions by Hairy Nanoparticles Bearing Polyanions"

_polymers, 2019, doi:10.3390/polym11050816_

Round 1

Reviewer 1 Report

Zhang et al. report on the preparation of core/shell particles consisting of polystyrene featuring a poly(sodium p-styrenesulfonate) shell by application of a starved feed emulsion polymerization protocol. The obtained particles were used as stabilizers for Pickering emulsions. The particles were investigated with respect to their capability to stabilize water/oil phases by means of optical microscopy and wettability by water contact angle measurements. The particles were investigated with respect to size, homogeneity and core/shell ratio. Moreover, the Pickering emulsions were prepared and investigated upon sonication treatment, standing time, salt additives and amounts of particles added. In general the manuscript is well written - needing some English polishing by a native speaker - the experiments and analytical data for all compounds are well described and can be found in the main manuscript or as part of the Supporting Information. This manuscript can potentially give a nice contribution to Polymers after taking the following critical points and suggestions into account (page numbers belong to the full pdf file sent):

Introduction: The authors should define the term “hairy”, which is from my point of view not a scientific one. Typically, polymer chemists define a grafting density for brushes. What is the grafting density and the brush thickness? Can the authors define or even avoid the term “hairy”, which is scientifically not correct and more jargon?

Minor: the “p” in “p-styrenesulfonate” should be italic.

Section 2.2.2: The citation for synthesizing PS@PSS is wrong. It is cited with ref 27, which belongs to the work by the Arms group aiming for PGMA-containing particles. I guess the authors wanted to cite ref 28 at this place.

Section 2.2.2: Please define the home-made reactor to be reproducible for the authors.

Page 5: The authors should be more precise and argument with the apparent difference of the average diameters by explaining the size of dried particles vs. the hydrodynamic diameter Rh of the particles in dispersion. The term Rh has not been mentioned in this case.

End of page 5: the effect of ultrasonication on Pickering emulsion preparation and stabilization is only given phenomenolgical. Is their any more detailed and more profound explanation (maybe by comparing with other systems known from literature) to underpin the authors‘ finding?

Figure 3 c: Resolution is poor, please improve.

Page 7, second paragraph: Can the authors give some exact values for shrinkage of the particles using higher ultrasonication intensities?

Page 7, lines 258-260: What might be a degradation mechanism? Which linkage will be broken upon ultrasonication?

Section 3.3 needs some English polishing. Reference 20 (line 294) is given by wrong citation style.

Author Response

Response to Reviewer 1 Comments

Point 1: Introduction: The authors should define the term “hairy”, which is from my point of view not a scientific one. Typically, polymer chemists define a grafting density for brushes. What is the grafting density and the brush thickness? Can the authors define or even avoid the term “hairy”, which is scientifically not correct and more jargon?

Response 1: Thanks for the reviewer’s suggestion. “Hairy particles” is composed of a core and a layer of polymer chains densely grafted via covalent bonds on the core surface, and this definition has been added in the last paragraph of introduction. Besides, photoemulsion polymerization was used to generate polymer chains and the grafting density could only be adjusted within certain limits. The grafting density of PS@PSS nanoparticles is near 0.05 cm-2 by calculation model which has been optimized in Guo’s work (Langmuir 2000, 16, 8719). The thickness of PS@PSS is 89.3 nm, which has been added in Table 1.

Point 2: Minor: the “p” in “p-styrenesulfonate” should be italic.

Response 2: We revised the format of “p” in “p-styrenesulfonate” in revised manuscript.

Point 3: Section 2.2.2: The citation for synthesizing PS@PSS is wrong. It is cited with ref 27, which belongs to the work by the Arms group aiming for PGMA-containing particles. I guess the authors wanted to cite ref 28 at this place.

Response 3: Very thanks for the reviewer’s careful check of our references. The citation has been changed to the right reference. Owing to the addition of other references, references number has been rearranged.

Point 4: Section 2.2.2: Please define the home-made reactor to be reproducible for the authors.

Response 4: Figure S1 in the “Supporting Information” gives the schematic diagram and photo of our home-made photo-reactor.

Point 5: Page 5: The authors should be more precise and argument with the apparent difference of the average diameters by explaining the size of dried particles vs. the hydrodynamic diameter Rh of the particles in dispersion. The term Rh has not been mentioned in this case.

Response 5: We have added the term Rh in the manuscript and the precise data of diameter difference between the hydrodynamic diameter (Rh) via DLS and dried particle size was given in the second paragraph of Section 3.1.

Point 6: End of page 5: the effect of ultrasonication on Pickering emulsion preparation and stabilization is only given phenomenolgical. Is their any more detailed and more profound explanation (maybe by comparing with other systems known from literature) to underpin the authors’ finding?

Response 6: Costa et al. also reported that zeta potential of chitosan nanoparticles gradually decreased and Pickering emulsion droplets became more homogeneous when ultrasonic power was tuned from 100 W to 450 W, but more drastic treatment (600 W) were accompanied by an increase in zeta potential. This comparison has been added in the second paragraph of Section 3.2.

Point 7: Figure 3 c: Resolution is poor, please improve.

Response 7: The resolution of Figure 3c has been improved in the revised manuscript.

Point 8: Page 7, second paragraph: Can the authors give some exact values for shrinkage of the particles using higher ultrasonication intensities?

Response 8: It has been added in line 248 that PS@PSS diameter decreased dramatically from 252 nm to 236 nm when ultrasonic power increased from 100 W to 300 W, which could also be found in Table 2.

Point 9: Page 7, lines 258-260: What might be a degradation mechanism? Which linkage will be broken upon ultrasonication?

Response 9: Changes of polymer structure can be created during ultrasonic treatment as a result of cavitation and research has suggested that chain cleavage could occur preferentially at weak spots in the chain (Annu. Rev. Mater. Sci. 1995, 20, 29]. PS@PSS hairy nanoparticles used here is a special fuzzy structure of dense core and loosened shell, so the thickness decrease of PS@PSS shown in Table 2 happened gradually from the chains outside to inner side.

Point 10: Section 3.3 needs some English polishing. Reference 20 (line 294) is given by wrong citation style.

Response 10: The citation style of ref 20 has been revised. Besides, we polished the language and some description in Section 3.3.

We thank for the reviewer’s useful comments and hope that our responses to the comments would satisfy him/her.

Reviewer 2 Report

Dear Authors,

the paper seems to be very interesting. Unfortunately, the description what was done is very incomplete. Recipes for the experiments [g] are very incomplete.  Thus, I was not able to follow what you have done. Furthermore, I miss experiments with varying amount of pickering particles. In the introduction “Conventional emulsions used in daily life are commonly stabilized by low molecular weight 32 surfactants that are thermodynamically stable. These generally only work at surfactant 33 concentrations exceeding the critical micelle concentration (CMC); however” is as formulated not true.

Kind regards

Author Response

Response to Reviewer 2 Comments

Point 1: The description what was done is very incomplete. Recipes for the experiments [g] are very incomplete.  Thus, I was not able to follow what you have done. Furthermore, I miss experiments with varying amount of pickering particles.

Response 1: Thanks for the reviewer’s suggestion on experiments. The detailed description of experiments was added in Chapter 2.2.2 and 2.3. Furthermore, the effect of PS@PSS nanoparticle concentration on the stabilization of Pickering emulsion was investigated in Chapter 3.7.

Point 2: Chapter 2.3 does not give any balance information. How does the homogenization device look like? How does the reactor look like? There is no geometry information.

Response 2: Figure S2 in “Supporting Information” gives the schematic diagram and photo of the homogenization device, which was called ultrasonic homogenizer.

Point 3: Chapter 2.4.1. Dilution is not described. How was the right dilution determined? What kind of PDI is used.

Response 3: The concentration of the fresh prepared PS@PSS is 1.02 wt% and we dilute it to 0.01 wt% with pure water. For example, 0.1 mL PS@PSS was diluted with 10 mL H2O. Besides, Gaussian polydispersity index (PDI) was used in the size analysis by DLS. All the information has been added in Chapter 2.4.1.

Point 4: As an example: For me it is absolutely not clear how the particles were generated which are pictured in Figure 3a.

Response 4: We’ve added the detailed experimental procedures in Chapter 2.3.

Point 5: In 2.2.2. the temperature information is missed.

Response 5: The experimental temperature is 25 °C and we added the information in Chapter 2.2.2.

Point 6: In the introduction “Conventional emulsions used in daily life are commonly stabilized by low molecular weight 32 surfactants that are thermodynamically stable. These generally only work at surfactant 33 concentrations exceeding the critical micelle concentration (CMC); however” is as formulated not true.

Response 6: Thanks very much and we accept the reviewer’s suggestion. We revised the description in the introduction.

We thank for the reviewer’s useful comments and hope that our responses to the comments would satisfy him/her.

Reviewer 3 Report

In this work, HS@PSS hairy particles were prepared with emulsion photopolymerization and surface-initiated polymerization. The prepared nanoparticles were used as stabilizers for preparation of Pickering emulsion. Several parameters were studied to explain the influence on performance of produced Pickering emulsion including ultrasonic power and time, standing time, oil phase types, and salt concentration. Very detailed experiments and discussion were presented, and this work provides good reference for stimuli-responsive Pickering emulsion preparation. Several questions need to be addressed:

1.      Stimuli-responsive is a key word in this work but the authors did not put enough effort on it instead just showing a hairy particles chemical structure. What kind of stimuli-responsiveness(es) these particles have? How do they influence the preparation and stabilization of Pickering emulsion? The application of stimuli-responsive emulsion needs the feasibility of preparation and breaking of emulsion, while the authors only focused on the preparation and stability in this manuscript. Conditions of breaking the emulsion is recommended to be included as well. But first, the authors need to clarify the scope of the particles’ stimuli-responsiveness(es).

2.      Page 9 line 290-294: The authors are correct with this statement. This partial wettability also has degree. There is supposed to be a suitable wettability helps to establish thermodynamic stability of the formed emulsion. pH would be a parameter to play with and achieve tunable wettability which was missed in this manuscript. The authors should do some experiments with pH influence on the preparation of Pickering emulsion with the hairy particles. Brush or arm density and length of the hairy particles also affect the wettability of hairy particles. If a real systemic study is expected as claimed in the introduction by the authors, more experiments are supposed to be completed and included in this manuscript.

Author Response

Response to Reviewer 3 Comments

Point 1: Stimuli-responsive is a key word in this work but the authors did not put enough effort on it instead just showing a hairy particles chemical structure. What kind of stimuli-responsiveness(es) these particles have? How do they influence the preparation and stabilization of Pickering emulsion? The application of stimuli-responsive emulsion needs the feasibility of preparation and breaking of emulsion, while the authors only focused on the preparation and stability in this manuscript. Conditions of breaking the emulsion is recommended to be included as well. But first, the authors need to clarify the scope of the particles’ stimuli-responsiveness(es).

Response 1: Thanks for the reviewer’s suggestion. Actually, the PS@PSS nanoparticles only have stimuli-responsiveness to salt concentration because of the electrostatic shielding effect, but the particles are not responsive to other stimuli. The stability of Pickering emulsion stabilized by PS@PSS nanoparticles could be controlled by changing salt concentration, which has been investigated in Section 3.5. Emulsion could exist stably for over 3 months when salt concentration was low (eg. 0.1 mM). However, once salt concentration increased, the additional ions shielded the electrostatic repulsive-force between PSS polymer chains, and the emulsion became unstable and gradually broke. So we can achieve the demulsification of Pickering emulsion and release the oil phase by increasing salt concentration.

Point 2: Page 9 line 290-294: The authors are correct with this statement. This partial wettability also has degree. There is supposed to be a suitable wettability helps to establish thermodynamic stability of the formed emulsion. pH would be a parameter to play with and achieve tunable wettability which was missed in this manuscript. The authors should do some experiments with pH influence on the preparation of Pickering emulsion with the hairy particles. Brush or arm density and length of the hairy particles also affect the wettability of hairy particles. If a real systemic study is expected as claimed in the introduction by the authors, more experiments are supposed to be completed and included in this manuscript.

Response 2: Thanks for the reviewer’s suggestion on experiments and agreement with our statement. Actually, PS@PSS nanoparticles are not responsive to pH. Table S1 and Figure S3 in the “Supporting Information” give the change of PS@PSS diameter with pH. But Pickering emulsions stabilized by other hairy particles which have pH-responsiveness are under investigation in our lab.

The photoemulsion polymerization was used to generate polymer chains as reported in our previous work and the grafting density could be rarely adjusted. The grafting density of PS@PSS nanoparticles is near 0.05 cm-2 by calculation model which has been optimized in Guo’s work (Langmuir 2000, 16, 8719). This data has been provided in Table 1. The effect of chain length on the emulsions is under study and would be published in our next paper.

Furthermore, in order to strengthen the systematicity of this study, the effect of PS@PSS nanoparticle concentration on the stabilization of Pickering emulsion was investigated in Chapter 3.7.

We thank for the reviewer’s useful comments and hope that our responses to the comments would satisfy him/her.

Round 2

Reviewer 2 Report

is OK

Reviewer 3 Report

Questions are well answered.